# Effects of calcium-containing phosphate binders on cardiovascular events and mortality in predialysis CKD stage 5 patients

**Ping-Huang Tsai**[1], **Chi-Hsiang Chung**[2,3], **Wu-Chien Chien**[2,4,5], **Pauling Chu**[1]*

**1** Division of Nephrology, Department of Internal Medicine, Tri-Service General Hospital, National Defense Medical Center, Taipei, Taiwan, **2** School of Public Health, National Defense Medical Center, Taipei, Taiwan, **3** Taiwanese Injury Prevention and Safety Promotion Association, Taipei, Taiwan, **4** Graduate Institute of Life Sciences, National Defense Medical Center, Taipei, Taiwan, **5** Department of Medical Research, Tri-Service General Hospital, Taipei, Taiwan

* pauling.chu@gmail.com

**Data Availability Statement:** All relevant data are within the manuscript and its Supporting Information files.

## Abstract

### Background

Hyperphosphatemia and calcium load were associated with vascular calcification. The role of calcium-containing phosphate binders (CCPBs) use as important determinants of death and cardiovascular events in predialysis hyperphosphatemic chronic kidney disease (CKD) patients remain unclear due to the absence of evidence for reduced mortality with CCPB use compared with placebo. We aimed to investigate the effect of using CCPBs or nonuse in all-cause mortality rates and cardiovascular events in CKD stage 5 patients between 2000 and 2005 in the Taiwanese National Health Insurance Research Database.

### Methods

Patients with known coronary heart disease and those who had undergone dialysis or renal transplantation were excluded. The CCPB users were matched with nonusers by the propensity score model. Multivariable Cox proportional hazards model was used to estimate hazard ratios (HRs) of all-cause mortality and cardiovascular events.

### Results

During a mean follow-up of 4.58 years, 879 CCPB users were matched with 3516 nonusers. CCPB use was an independent risk factor for cardiovascular events [adjusted hazard ratio (HR) 1.583, 95% confidence interval (CI) 1.393–1.799]. The increased cardiovascular risk was dose-dependent and consistent across all subgroup analyses. Compared with no use, CCPB use was associated with no significant all-cause mortality risk (1.74 vs. 1.75 events per 100 person-years, adjusted HR 0.964, 95% CI 0.692–1.310).

### Conclusions

CCPB use in CKD stage 5 patients was associated with a significantly increased cardiovascular event risk compared with the nonusers, whereas the all-cause mortality risk was

**Funding:** The author(s) received no specific funding for this work.

**Competing interests:** The authors have declared that no competing interests exist.

similar between the two groups. Whether these relationships are causal require further randomized controlled trials.

## Introduction

Chronic kidney disease (CKD) is a worldwide public health issue that caused premature mortality or 20 million disability-adjusted life years in 2010 [1]. Accelerated and progressive vascular calcification might play major roles in the development of cardiovascular disease, the leading cause of death in CKD [2]. During the evolution of CKD, adaptive responses fail to maintain the calcium–phosphate homeostasis. Hyperphosphatemia develops gradually, with the prevalence increasing up to 40% in those with estimated glomerular filtration < 20 ml/min/1.73m [3, 4], which is associated with the progression of secondary hyperparathyroidism and renal bone disease. Observational studies showed a significant association between hyperphosphatemia and high mortality among patients on hemodialysis [2]. Furthermore, high serum phosphorus levels were shown to be associated with increased cardiovascular mortality and hospitalization risks among patients on hemodialysis [3]. As an inevitable clinical result of stage 5 CKD, controlling elevated serum phosphate levels by dietary restriction and intestinal phosphate binders is an important issue in clinical practice. Phosphate binders are suggested by the Kidney Disease: Improving Global Outcomes (KDIGO) clinical practice guidelines to treat hyperphosphatemia in patients with stage 3–5 nondialysis CKD [5].

Calcium-containing phosphate binders (CCPBs) reduce serum phosphorus levels in advanced CKD. However, the positive calcium balance observed in several studies suggested soft-tissue deposition after calcium exposure, implicating the potentially harmful effects of generous calcium supplementation in patients with stage 3–4 CKD and normal phosphate levels [6, 7]. CCPBs is an important cause of hypercalcemia in CKD population, raising the risk of vascular calcification and leading to cardiovascular disease [8]. The prevalence of hypercalcemia in CKD stage 4 and 5 patients was 13.4%, which was lower than dialysis patients. The cause of higher rate of hypercalcemia in dialysis patients might be due to greater usage of vitamin D analogues and high-dose CCPBs [9]. In hyperphosphatemic patients with predialysis CKD, CCPBs were reported to be associated with worse adverse effects on vascular calcification and significantly higher all-cause mortality compared with calcium-free binders [10, 11]. These results were reinforced by several systematic reviews [12–15], altogether suggesting that calcium-free phosphate binders might be either beneficial or otherwise not harmful compared with CCPBs in treating hyperphosphatemia in CKD. Based on the cumulative outcomes suggesting potential damage with CCPBs, the KDIGO CKD mineral bone disorder (MBD) guidelines recommend restricting the CCPB dose in predialysis patients with CKD [16].

It remains uncertain whether preventing the development of hyperphosphatemia with CCPBs might prevent or enhance the development of vascular calcification, cardiovascular risks and mortality and in patients with stage 5 CKD. Although CCPBs are used widely in patients with high cardiovascular risk, there are no adequately powered, well-designed, and long-term randomized controlled trials to evaluate whether regular CCPB use in stage 5 CKD affects serious outcomes such as mortality or cardiovascular events compared with placebo or no treatment. However, there is inadequate motivation for the pharmaceutical industry to support clinical trials for medications that are already considered necessary by practicing physicians and have attained high utility in the CKD population. We, therefore, conducted a nationwide population-based retrospective observational cohort study using the National

Health Insurance (NHI) Research Database (NHIRD) in Taiwan, which provided an exceptional opportunity to evaluate the effects of CCPBs on mortality and coronary heart disease risks in stage 5 CKD.

## Materials and methods

### Data source

This nationwide population-based cohort study uses Taiwan's NHIRD, comprising the standard health care data submitted by medical institutions seeking reimbursement through the NHI, which covers the medical needs of 99.19% of the 23 million individuals in the population of Taiwan. The identified information derived from the NHIRD included birth date, sex, residence area, drug prescriptions, medical procedures, and diagnostic codes according to the International Classification of Diseases 9th Revision-Clinical Modification (ICD-9-CM). The study was approved by the Institutional Review Board of Tri-Service General Hospital (TSGH IRB No. 2–105–05–082.).

### Study design and participants

This study included patients with a primary diagnosis of CKD who received erythropoiesis-stimulating agents (ESAs) between January 1, 2000, and June 30, 2005, and were followed up until December 31, 2005, as potential study subjects who were identified in the NHIRD.

The NHI reimbursement regulations in Taiwan [17] state that the ESA treatment can be initiated in predialysis patients with CKD accompanied with a serum creatinine concentration > 530 μmol/L (approximately equivalent to stage 5 CKD) and a hematocrit level < 28% to maintain a packed-cell volume of no more than 36%. The selected cohort, which was validated previously, was a representative of the predialysis patients with stage 5 CKD in Taiwan [18, 19]. The current study included patients with a primary diagnosis of CKD (ICD-9 codes, including 016.0, 042, 095.4, 189, 223, 236.9, 250.4, 271.4, 274.1, 440.1, 442.1, 446.21, 447.3, 572.4, 580–589, 590–591, 593, 642.1, 646.2, 753, and 984) and those who were receiving ESAs covered by the NHI (identified by a serum creatinine concentration > 530 μmol/L). Patients younger than 18 years, those with coronary heart disease diagnosed before the study period and those who had undergone dialysis or renal transplantation before the ESA treatment, were excluded. The prescription information of the 90 days after the first ESA treatment was used to ascertain CCPB use, and the 91st day after the first ESA prescription was defined as the index date for study entry. The patients who commenced renal replacement therapy or died and those who were prescribed a CCPB between the first ESA treatment and the index date were also excluded.

The patients who were using a CCPB within the 90 days after the first ESA prescription were defined as CCPB users and the remaining patients were designated as nonusers. The ICD-9-CM codes were used to identify comorbidities that were diagnosed within the three years before the index date and included the following diagnoses (ICD-9-CM codes) that are considered to increase the mortality risk: diabetes mellitus (250), hypertensive cardiovascular disease (401–405), heart failure (428), cerebrovascular disease (430–438), cirrhosis (571), peripheral arterial occlusive disease (440.0, 440.2, 440.3, 440.8, 440.9, 443, 444.0, 444.22, 444.8, 447.8, and 447.9), cancer (140–208), and dyslipidemia (272). The Charlson comorbidity index was used to quantify the comorbidity profiles. To balance the two groups based on known confounding factors, the propensity score for the likelihood of receiving CCPBs was calculated using the baseline covariates listed in Table 1. The CCPB users were matched to the nonusers using a ratio of 1:4 based on age, sex, and the propensity score.

**Table 1. Baseline characteristics of the study patients by calcium-containing phosphate binder use before and after propensity score matching.**

| | Before propensity score matching | | | After propensity score matching | | |
|---|---|---|---|---|---|---|
| | User (n = 879) | Nonuser (n = 7245) | *P* value | User (n = 879) | Nonuser (n = 3516) | *P* value* |
| Age (years) | 64.80 ± 13.72 | 60.75 ± 14.96 | <0.001 | 64.80 ± 13.72 | 64.63 ± 14.39 | 0.751 |
| Age group (years) | | | 0.025 | | | 0.999 |
| 18–44 | 87 (9.90) | 806 (11.12) | | 87 (9.90) | 348 (9.90) | |
| 45–64 | 308 (35.04) | 2,801 (38.66) | | 308 (35.04) | 1,232 (35.04) | |
| ≥65 | 484 (55.06) | 3,638 (50.21) | | 484 (55.06) | 1,936 (55.06) | |
| Sex | | | 0.150 | | | 0.999 |
| Male | 446 (50.74) | 3,862 (53.31) | | 446 (50.74) | 1,784 (50.74) | |
| Female | 433 (49.26) | 3,383 (46.69) | | 433 (49.26) | 1,732 (49.26) | |
| Comorbidities | | | | | | |
| Diabetes mellitus | 322 (36.63) | 2,624 (36.22) | 0.809 | 322 (36.63) | 1,295 (36.83) | 0.913 |
| Hypertensive cardiovascular disease | 326 (37.09) | 1,949 (26.90) | <0.001 | 326 (37.09) | 1,277 (36.32) | 0.672 |
| Dyslipidemia | 24 (2.73) | 297 (4.10) | 0.049 | 24 (2.73) | 89 (2.53) | 0.739 |
| Cirrhosis | 29 (3.30) | 582 (8.03) | <0.001 | 29 (3.30) | 125 (3.56) | 0.712 |
| Cancer | 53 (6.03) | 487 (6.72) | 0.436 | 53 (6.03) | 215 (6.11) | 0.925 |
| Cerebrovascular disease | 53 (6.03) | 377 (5.20) | 0.302 | 53 (6.03) | 217 (6.17) | 0.875 |
| Heart failure | 100 (11.38) | 894 (12.34) | 0.441 | 100 (11.38) | 397 (11.29) | 0.943 |
| Charlson comorbidity index score | 0.18 ± 0.50 | 0.28 ± 0.63 | <0.001 | 0.18 ± 0.50 | 0.21 ± 0.55 | 0.123 |

* The chi-square or Fisher's exact test was used to evaluate categorical variables, and Student's *t*-test was used for continuous variables.

## Study outcomes and follow-up

The primary outcome was all-cause mortality. The primary diagnosis of in-hospital death or the first-listed discharge diagnosis at the last hospitalization within the three months before death for out-of-hospital deaths was defined as the cause of death, as indicated previously [20]. To identify the association between CCPBs and coronary artery calcification in patients with CKD, the number of admissions for coronary heart disease (ICD-9 codes 410–414, excluding 412 and 414.1) was also determined.

## Statistical analysis

The clinical characteristics of the CCPB users and nonusers were described using percentages and means ± standard deviation for categorical and continuous variables, respectively. Differences between the CCPB users and nonusers were compared by independent Student's *t*-test or the χ2 test, where appropriate. The cumulative incidences of all-cause mortality and coronary heart disease over time were compared between the CCPB users and nonusers using the Kaplan–Meier method.

The multivariable Cox proportional hazards model was used before and after propensity score matching to estimate hazard ratios (HRs) of outcomes after controlling for age, sex, diabetes mellitus, hypertensive cardiovascular disease, heart failure, cerebrovascular disease, cirrhosis, cancer, dyslipidemia, and Charlson comorbidity index score. Subgroup analyses were also performed to determine the HRs of the outcomes in the CCPB users and nonusers.

To assess cumulative dose-related effects, the risks of mortality and coronary heart disease were evaluated according to the defined daily dose (DDD) during the 90-day exposure period (≤15, 16–40, and >40 DDD) and the prescribed daily dose (≤500, 501–1000, and >1000 mg) relative to no CCPB use. DDD represents as a technical unit of measurement which defines the assumed usual maintenance dose per day for a drug used for its main indication in adults.

Though only give an estimate of consumption and not an exact picture of actual use, DDD is useful in assessing trends in drug consumption and performing comparisons between population groups. A two-tailed $P$ value $< 0.05$ was considered statistically significant. All statistical analyses were performed using SAS (version 9.3; SAS Institute) and Stata SE (version 11.0; StataCorp).

## Results

### Patient characteristics

A total of 8124 patients diagnosed with stage 5 CKD between January 1, 2000, and June 30, 2005, including 879 CCPB users and 7245 nonusers, comprised the study cohort, as depicted in Fig 1. Table 1 shows the baseline characteristics of the two groups categorized by CCPB use before and after propensity score matching. Before matching, the users were older and more likely to have hypertensive cardiovascular disease. The rates of those with cirrhosis and higher Charlson comorbidity index scores were greater among the nonusers compared with the CCPB users. After matching with a 1:4 ratio according to the propensity score, 879 CCPB users and 3516 nonusers were included in the primary outcome; the baseline characteristics of the two groups after matching were comparable.

### The association of CCPB use with treatment outcomes

During the follow-up period, coronary heart disease was diagnosed in 462/879 (52.55%) CCPB users and 1367/3516 (38.87%) nonusers. The incidence of coronary heart disease was higher

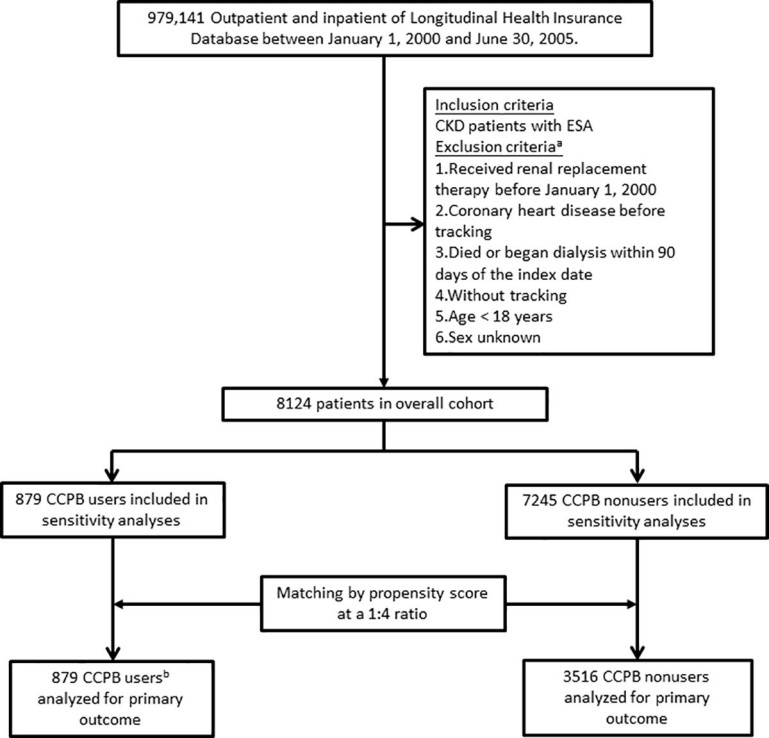

**Fig 1. Study profile.** [a]Numbers for exclusions may not sum because of patients fulfilling more than one criterion. [b]Patients in the treated cohort (received CCPB for $\geq$ 90 days) were matched at a ratio of 1:4 with those in the untreated cohort (never received CCPB) by means of propensity scores. Abbreviations: CKD, chronic kidney disease; ESA, erythropoiesis-stimulating agent; CCPB, calcium-containing phosphate binder.

among the CCPB users compared with the nonusers (11.99 vs. 8.27 per 100 person-years, crude HR 1.607, 95% confidence interval [CI] 1.442–1.891). After adjustment for age, sex, baseline comorbidities, and Charlson comorbidity index score, the risk for coronary heart disease remained higher in the CCPB users (adjusted HR 1.583, 95% CI 1.393–1.799) (Table 2). However, the risk for all-cause mortality was not significantly different between the CCPB users and the nonusers. In the matched cohort, 111/879 (12.6%) CCPB users and 333/3516 (9.47%) nonusers died during the follow-up period (incidence, 1.74 and 1.75 events per 100 person-years for the CCPB users and the nonusers, respectively; crude HR 0.896, 95% CI 0.521–1.112) (Table 2). The association did not change after adjusting for the baseline covariates (adjusted HR 0.964, 95% CI 0.692–1.310).

The apparent adverse effects of CCPBs increased substantially with increasing doses. Compared with the nonusers, those using CCPBs had an increased risk of coronary heart disease, and those who were using CCPBs at more than 1000 mg/day had the greatest risk (adjusted HR 1.73, 95%CI 1.518–1.962; $P = 0.038$ for trend). An increased risk of coronary heart disease was noted with the cumulative CCPB doses greater than 40 DDD (adjusted HR 1.649, 95% CI 1.444–1.878; $P = 0.064$ for trend). However, this dose-response relationship was not noted for all-cause mortality (Table 2).

For coronary heart disease and all-cause mortality, Kaplan–Meier method was used. Events during the first 3 months were excluded. The cumulative incidence for coronary heart disease was significantly higher among the CCPB users compared with the nonusers (Fig 2A); however, a similar difference was not observed for all-cause mortality (Fig 2B).

## Subgroup and sensitivity analyses

The subgroup and sensitivity analyses were consistent with the analyses of the two main groups, with a comparison of the treated (received CCPB for $\geq$ 90 days) and untreated (never received CCPB) cohorts. Among patients with advanced CKD, CCPB use was found to be associated with increased risk of coronary heart disease across all clinically relevant subgroups (stratified by age, sex, diabetes mellitus, hypertensive disease, dyslipidemia, liver cirrhosis,

**Table 2. Risks of all-cause mortality and coronary heart disease in predialysis patients with chronic kidney disease after propensity score matching (n = 4395).**

| | All-cause mortality | | | | Coronary heart disease | | | |
|---|---|---|---|---|---|---|---|---|
| | Events | Incidence | Crude HR | Adjusted HR | Events | Incidence | Crude HR | Adjusted HR |
| | (n/N) | (per 100 PYs) | (95% CI) | (95% CI) | (n/N) | (per 100 PYs) | (95% CI) | (95% CI) |
| **Matched cohort** | | | | | | | | |
| Nonusers | 333/3516 | 1.75 | 1 | 1 | 1367/3516 | 8.27 | 1 | 1 |
| Users | 111/879 | 1.74 | 0.896 (0.521–1.112) | 0.964 (0.692–1.310) | 462/879 | 11.99 | 1.607 (1.442–1.891) | 1.583 (1.393–1.799) |
| **Defined daily dose** | | | | | | | | |
| $\leq$15 DDD | 33/276 | 1.82 | 0.938 (0.542–1.163) | 1.006 (0.721–1.368) | 121/276 | 11.31 | 1.516 (1.358–1.790) | 1.492 (1.311–1.699) |
| 16–40 DDD | 39/291 | 1.79 | 0.924 (0.531–1.145) | 0.995 (0.709–1.352) | 153/291 | 11.98 | 1.608 (1.440–1.892) | 1.585 (1.390–1.801) |
| $\geq$41 DDD | 39/312 | 1.64 | 0.845 (0.488–1.049) | 0.903 (0.648–1.234) | 188/312 | 12.47 | 1.673 (1.498–1.964) | 1.649 (1.444–1.878) |
| *P* for trend | | | | 0.279 | | | | 0.064 |
| **Prescribed daily dose** | | | | | | | | |
| $\leq$500 mg/day | 19/183 | 1.96 | 1.009 (0.583–1.251) | 1.084 (0.777–1.483) | 89/183 | 10.24 | 1.372 (1.231–1.631) | 1.352 (1.184–1.538) |
| 501–1000 mg/day | 34/279 | 1.83 | 0.937 (0.542–1.167) | 1.011 (0.718–1.386) | 124/279 | 11.45 | 1.531 (1.376–1.813) | 1.511 (1.326–1.724) |
| $\geq$1000 mg/day | 58/417 | 1.64 | 0.848 (0.489–1.052) | 0.91 (0.651–1.240) | 249/417 | 13.09 | 1.759 (1.574–2.084) | 1.73(1.518–1.962) |
| *P* for trend | | | | 0.156 | | | | 0.038 |

Abbreviations: CI, confidence interval; DDD, defined daily dose; HR, hazard ratio; PYs, person-years.

**(A)** Coronary heart disease

**(B)** All-cause mortality

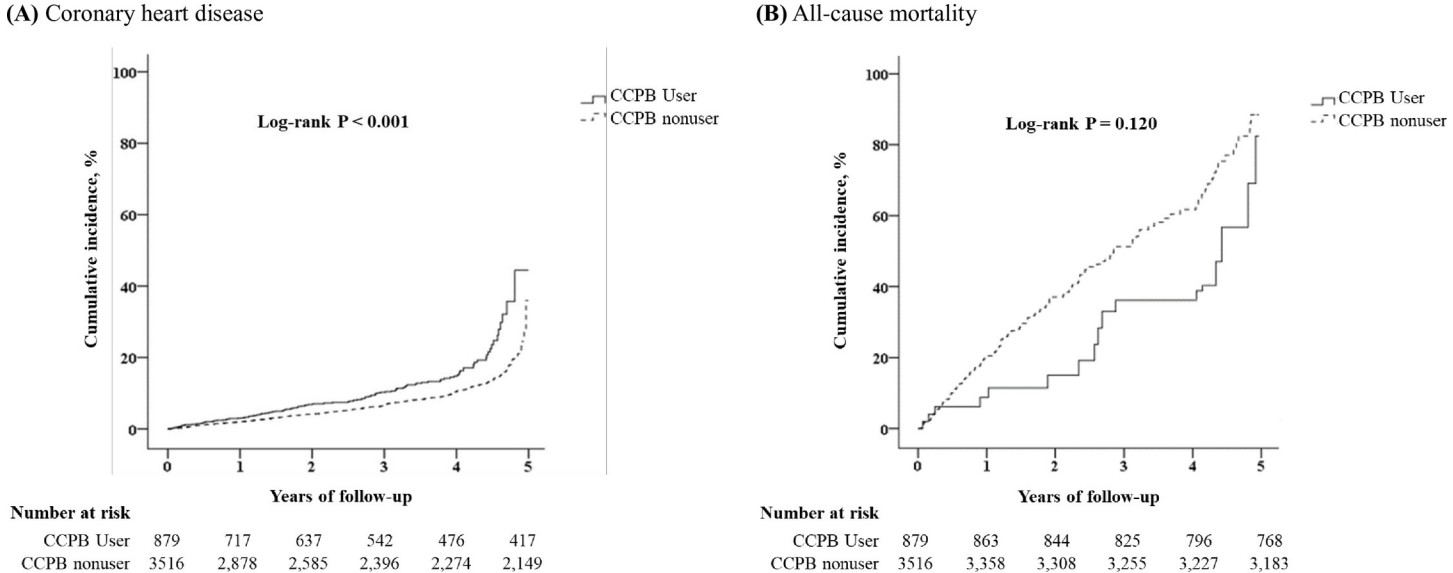

**Fig 2.** Kaplan–Meier Curves for the Cumulative Incidence of (**A**) Coronary Heart Disease and (**B**) All-cause Mortality with Calcium-containing Phosphate Binder Use. Comparison of the cumulative incidence of coronary heart disease and all-cause mortality were shown. *P* values<0.05 were considered statistically significant.

cancer, cerebrovascular disease and heart failure) (Fig 3A), whereas the all-cause mortality risk was not significant in any of the subgroup analyses (Fig 3B).

## Discussion

This national population-based observational cohort study showed that CCPB use was associated with a 58% increased risk of coronary heart disease among patients with CKD stage 5 after the propensity score matching. A higher risk was noted in those on higher CCPB doses; this result was supported by the observed positive dose-response relationship. Furthermore, these findings were confirmed with sensitivity and subgroup analyses. Although the outcomes of observational studies should be interpreted carefully, the observed dose-dependent association provides strong support for a true relationship between CCPBs and coronary heart disease in advanced CKD. Another major study finding was the lack of an association between CCPBs and higher all-cause mortality risk in predialysis patients with CKD.

Hyperphosphatemia is associated with a higher risk of vascular calcification and worse cardiovascular outcomes [21–25]. CCPBs are widely used to control serum phosphate levels; however, concerns remain regarding their effect on the calcium balance. The risk of hypercalcemia was higher in the CCPB users compared with nonusers [14, 26]. In the CKD population, the harmful effects of calcium are consistent with the role of calcium in the pathophysiology of vascular calcification [27, 28], which may contribute to the initiation or progression of vascular stiffness [29] and coronary artery calcification [30]. In the current study, we found an increase in coronary heart disease in the patients on CCPBs compared with the nonusers, which lends further evidence for the contribution of the positive calcium balance to vascular calcification. Similarly, we found an increased coronary heart disease risk with higher cumulative CCPB doses. The progression of coronary artery calcification was demonstrated to be a reliable method to predict mortality [31, 32], which support the theoretical risk of inducing a positive calcium balance in patients with CKD, in whom an abnormal mineral metabolism might result in dystrophic calcification and higher mortality. To date, two observational studies have examined the association of phosphate binders with a survival benefit in predialysis CKD patients

**(A)** Coronary heart disease

**A**

| Characteristic | | User | Nonuser | Hazard ratio (95% CI) |
| --- | --- | --- | --- | --- |
| Gender | Male | 232 | 604 | 1.951 (1.58–2.409) |
| | Female | 230 | 763 | 1.577 (1.278–1.946) |
| Age group (years) | 18-44 | 31 | 78 | 1.439 (1.121–1.828) |
| | 45-64 | 168 | 556 | 1.565 (1.177–2.046) |
| | ≥ 65 | 263 | 733 | 1.845 (1.112–3.062) |
| Diabetes mellitus | With | 208 | 671 | 1.697 (1.317–2.185) |
| | Without | 254 | 696 | 1.337 (1.025–1.721) |
| Hypertensive disease | With | 207 | 530 | 1.769 (1.468–2.131) |
| | Without | 255 | 837 | 1.375 (1.061–1.782) |
| Dyslipidemia | With | 21 | 78 | 1.767 (1.519–2.055) |
| | Without | 441 | 1,289 | 0.987 (0.252–3.864) |
| Liver cirrhosis | With | 11 | 43 | 2.354 (1.481–4.596) |
| | Without | 451 | 1,324 | 1.433 (1.203–1.701) |
| Cancer | With | 13 | 41 | 2.777 (1.672–4.821) |
| | Without | 449 | 1,326 | 1.458 (1.326–1.806) |
| Cerebrovascular disease | With | 23 | 135 | 1.761 (1.509–2.054) |
| | Without | 439 | 1,232 | 1.346 (0.751–2.411) |
| Heart failure | With | 75 | 233 | 2.112 (1.287–3.463) |
| | Without | 387 | 1,134 | 1.528 (1.475–1.724) |
| | **Overall** | 462 | 1,367 | 1.583 (1.393–1.799) |

Abbreviation: CI, confidence interval

**(B)** All-cause Mortality

**B**

| Characteristic | | User | Nonuser | Hazard ratio (95% CI) |
| --- | --- | --- | --- | --- |
| Gender | Male | 63 | 187 | 0.97 (0.648–1.351) |
| | Female | 48 | 146 | 0.868 (0.652–1.257) |
| Age group (years) | 18-44 | 7 | 31 | 0.648 (0.436–1.062) |
| | 45-64 | 32 | 129 | 0.876 (0.386–1.992) |
| | ≥ 65 | 72 | 173 | 1.064 (0.807–1.402) |
| Diabetes mellitus | With | 27 | 102 | 0.967 (0.744–1.43) |
| | Without | 84 | 231 | 0.723 (0.471–1.108) |
| Hypertensive disease | With | 31 | 47 | 1.486 (0.943–2.342) |
| | Without | 80 | 286 | 0.818 (0.637–1.05) |
| Dyslipidemia | With | 0 | 0 | - |
| | Without | 111 | 333 | 0.964 (0.692–1.31) |
| Liver cirrhosis | With | 5 | 36 | 1.024 (0.739–1.515) |
| | Without | 106 | 297 | 0.622 (0.218–1.267) |
| Cancer | With | 14 | 85 | 0.997 (0.787–1.314) |
| | Without | 97 | 248 | 0.743 (0.42–1.262) |
| Cerebrovascular disease | With | 14 | 36 | 1.491 (0.789–2.816) |
| | Without | 97 | 297 | 0.851 (0.676–1.071) |
| Heart failure | With | 11 | 32 | 1.038 (0.719–1.689) |
| | Without | 100 | 301 | 0.902 (0.416–1.133) |
| | **Overall** | 111 | 333 | 0.964 (0.692–1.31) |

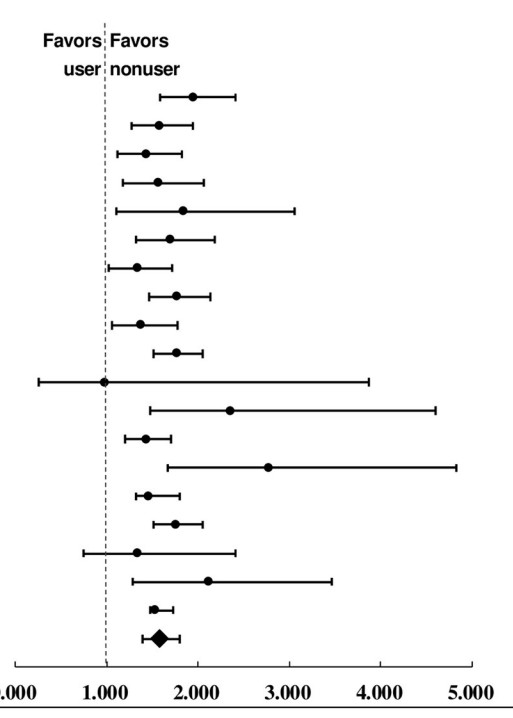

Abbreviation: CI, confidence interval

**Fig 3.** Multivariable Stratified Analyses of the Association Between Calcium-containing Phosphate Binder Use and (**A**) Coronary Heart Disease Development and (**B**) All-cause Mortality. Hazard ratios from the subgroup analysis for coronary heart disease and all-cause mortality between CCPBs user and nonuser were shown.

[33, 34], with different results of phosphate binder use. In addition to selection bias, sex, and different CKD stages, the heterogeneity of the phosphate binders use might have also contributed to the conflicting results regarding the survival benefit. In the current study, we focused on the effect of CCPBs on vascular events and mortality and therefore enrolled patients during a period of time when phosphate binders were generally restricted to CCPBs in Taiwan, with consideration of confounding factors such as marketing efforts in promoting non-CCPBs [35]. However, we were unable to demonstrate whether the increase in coronary heart disease in CCPB users was due to an increase in the positive calcium balance or phosphate toxicity because the relevant laboratory values, including baseline calcium, phosphorus, and intact parathyroid hormone concentrations, were not available for adjustments or comparisons. Therefore, well-designed placebo-controlled trials are warranted to determine whether CCPB use in predialysis CKD patients obtain more benefits from lowering phosphate levels or harm in increasing calcium levels.

Winkelmayer *et al.* evaluated whether CCPBs were associated with increased mortality risk in dialysis-dependent patients with CKD and found no association between CCPB use and 1-year all-cause mortality in dialysis patients (adjusted HR for all-cause mortality in users vs. nonusers 0.85, 95% CI 0.72–1.10) [36]. Although no significant reduction in mortality was observed in the study by Winkelmayer *et al.* and our study, several differences between these two studies merit discussion. While Winkelmayer *et al.* enrolled patients on dialysis, we included only predialysis patients. To determine whether lowering serum phosphate with CCPBs achieves net benefits or risks, an association of lower serum phosphate levels with reduced mortality and cardiovascular event rates should be demonstrated. However, dialysis removes phosphorus more efficiently than CCPBs. Thus, we could not distinguish whether the benefit of lowering phosphate was attributable to the phosphate binders or phosphorus removal via dialysis. In the current study, we limited the entire cohort to patients with advanced CKD not on dialysis to remove the confounding effect of dialysis on phosphate levels. Furthermore, we investigated the effect of CCPBs on cardiovascular events, which was not included in the study by Winkelmayer *et al.* Another difference between the studies is the observation time, which was longer in the current study.

Several important strengths of our study merit discussion. Although clinical trials are the gold standard for evaluating the comparative efficacy of therapeutic interventions, observational studies are useful to evaluate the benefits and adverse effects of CCPBs in the absence of well-designed randomized controlled trials evaluating CCPB versus placebo in predialysis patients with CKD. The study cohort was a large national representation of CCPB use to assess its effect on the development of coronary heart events in predialysis patients with CKD, which is a clinically significant issue that, to our knowledge, has not yet been addressed. Furthermore, the current study examined the effect of CCPB use on all-cause mortality in these patients. Our findings were robust across sensitivity and subgroup analyses, indicating that these findings are generalizable to the predialysis CKD populations with multiple concomitant comorbidities. However, several limitations of the current study should also be acknowledged. First, use and nonuse of CCPB were determined within 90 days after the index date; however, the drug status was not ascertained during the follow-up period and misclassification of drug exposure of interest is possible, which could have resulted in the observed null association between CCPB use and all-cause mortality. And the use of DDD did not permit a more meaningful comparison of calcium exposure as compared to the accurate computation of daily elemental calcium received from phosphate binders. Second, the unmatched CCPB nonusers were generally younger and the rate of hypertension, a known cardiovascular risk factor, was lower, compared with the CCPB users matched by the propensity score. Although the analyses were adjusted for demographic and patient-level clinical variables, this was an observational

study with potential unmeasured confounders, such as alcohol or tobacco use, other medication use including lanthanum, sevalamer, warfarin or active vitamin D treatment, and echocardiographic abnormalities. Besides, the laboratory data were not included, and whether baseline phosphorus, calcium, CRP, and parathyroid hormone levels were comparable between the two groups remains uncertain. Furthermore, data on baseline creatinine levels and estimated glomerular filtration rates were not available. The intrinsic limitation of the NHIRD did not allow us to explore a larger CKD cohort of patients with creatinine concentrations < 530 μmol/L who could be identified due to the NHI regulations on ESA prescriptions. Third, based on the different phosphorus content of each meal, a distinctive approach to CCPB use allows for patient authorization with self-adjustment of phosphate binders. Moreover, adherence to phosphate binders confounds the relationship between the medication prescription and the clinical outcomes. Fourth, for most patients, coronary artery calcification is not clinically apparent for at least ten years of dialysis treatment [37]. The duration of the current cohort study might be relatively short to provide solid evidence on the effects of CCPB treatments on vascular calcification or cardiovascular events. Finally, the current study included Taiwanese patients with advanced CKD not on dialysis, and the conclusions may not be extrapolated to other ethnicities or populations.

## Conclusions

In summary, compared with the nonusers matched by propensity score, CCPB users had a significantly higher coronary heart disease risk in the current national registry study of patients with stage 5 CKD. The increased coronary heart disease risk was dose-dependent and consistent across all subgroups of interest. However, CCPB use was not associated with higher all-cause mortality risk. Although insufficient in providing a useful cost-effectiveness model, the current study findings have important therapeutic implications, supporting the current KDIGO CKD-MBD recommendations and providing evidence that liberal CCPB use should be restricted in predialysis patients with CKD.

## Supporting information

**S1 File.**
(XLS)

## Acknowledgments

The authors thank enago academy (www.enago.tw) for writing assistance, technical editing and language editing.

## Author Contributions

**Conceptualization:** Pauling Chu.

**Data curation:** Chi-Hsiang Chung, Wu-Chien Chien.

**Formal analysis:** Ping-Huang Tsai, Wu-Chien Chien, Pauling Chu.

**Investigation:** Ping-Huang Tsai, Wu-Chien Chien.

**Methodology:** Ping-Huang Tsai, Wu-Chien Chien.

**Resources:** Wu-Chien Chien.

**Software:** Chi-Hsiang Chung.

**Validation:** Chi-Hsiang Chung.

**Visualization:** Chi-Hsiang Chung.

**Writing – original draft:** Ping-Huang Tsai.

**Writing – review & editing:** Wu-Chien Chien, Pauling Chu.

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
