## [Decision Letter · Decision Letter 0]

9 Jul 2020

PONE-D-20-04452

Effects of Calcium-Containing Phosphate Binders on Cardiovascular Events and Mortality in Predialysis CKD Stage 5 Patients

PLOS ONE

Dear Dr. Chu,

Thank you for submitting your manuscript to PLOS ONE. After careful consideration, we feel that it has merit but does not fully meet PLOS ONE’s publication criteria as it currently stands. Therefore, we invite you to submit a revised version of the manuscript that addresses the points raised during the review process.

SPECIFIC ACADEMIC EDITOR COMMENTS: Four expert reviewers in the field handled your manuscript. We thank them for their time and efforts. Although interest was found in your study, there were major concerns that arose during review that overshadowed this enthusiasm. These concerns include the need to better explain several vague comments, including the rationale for conducting this study; questions about the experimental design, including dose and duration of CCPB versus non-calcium based phosphate binder usage; more specifics about the patients cohort need to be provided and additional outcomes are requested; and there are comments about the limitations of this study. All reviewers' comments must be addressed in the revised manuscript.

We look forward to receiving your revised manuscript.

Kind regards,

Frank T. Spradley

Academic Editor

PLOS ONE

2. Please upload a new copy of Figures 1 & 3 as the detail is not clear. Please follow the link for more information: https://blogs.plos.org/plos/2019/06/looking-good-tips-for-creating-your-plos-figures-graphics/" https://blogs.plos.org/plos/2019/06/looking-good-tips-for-creating-your-plos-figures-graphics/

Reviewers' comments:

Reviewer's Responses to Questions

**Comments to the Author**

1. Is the manuscript technically sound, and do the data support the conclusions?

Reviewer #1: Yes

Reviewer #2: Yes

Reviewer #3: Yes

Reviewer #4: Yes

2. Has the statistical analysis been performed appropriately and rigorously? 

Reviewer #1: Yes

Reviewer #2: Yes

Reviewer #3: Yes

Reviewer #4: Yes

3. Have the authors made all data underlying the findings in their manuscript fully available?

Reviewer #1: Yes

Reviewer #2: Yes

Reviewer #3: Yes

Reviewer #4: Yes

4. Is the manuscript presented in an intelligible fashion and written in standard English?

Reviewer #1: Yes

Reviewer #2: Yes

Reviewer #3: Yes

Reviewer #4: Yes

5. Review Comments to the Author

Reviewer #1: The authors did an excellent job analyzing the effect of calcium containing phosphate binders on cardiovascular mortality.

However it is important that the authors identify with more detail the limitations of the study. The authors have mentioned the difficulty to separate Phosphate levels vs calcium load from the binder. There are other factors that should be commented: anticuoagulants-warfarin, treatment with active vitamin D, patients adherence to treatment not only relative to phosphtae control but also interdialytic weight gain, blood pressure medication, PTH levels , CRP levels and others

Reviewer #2: Dear Prof Spradley,

Re: Manuscript ("Effects of Calcium-Containing Phosphate Binders on Cardiovascular Events and Mortality in Predialysis CKD Stage 5 Patients" (PONE-D-20-04452))

Thank you for the kind invitation to review this manuscript. This study is one of the largest study to evaluate the relationship between calcium based phosphate binder and cardiovascular risk among pre-dialysis patients which has not been adequately examined. The study results are interesting and the manuscript is generally well-written.

Attached below are my comments and clarifications regarding the manuscript for the authors’ consideration.

Major comments

1) It will be good for the authors to comment on their focus on Stage 5 predialysis patients and also why patients in stage 4 were not considered. [Noted this was only brought up in the discussion briefly but may be good to describe in the methodology]

2) An important issue of note to evaluate the dose dependent relationship of calcium based phosphate binders is that the dose of elemental calcium within the CCB should be computed. This will facilitate more accurate and meaningful evaluation of this relationship as different CCPB have varying contents of elemental calcium. From the described methodology, it seems that it would be possible to compute the elemental calcium as the dose is available and the name of the CCPB.

3) How was the segregation of the subgroups of daily defined dose (≤15, 16–40, and >40 DDD) determined?

4) An important factor for consideration that would substantially strengthen the relationship between use of CCPB and adverse outcomes would be the use of non-calcium based phosphate binders e.g. lanthanum, and sevalamer as well as vitamin D analogues. It will be good to report these information among users of CCPB and non-users and adjust for theses confounders (especially non-calcium based phosphate binders) in the multivariate analyses.

5) Discussion: beyond the limits of study designs (comparisons made between winkelmayer et al), are there any potential reasons for the finding that there were no relationship between use of calcium based phosphate binders and mortality? In general most renal physicians will endeavour to avoid hypercalcemia among pre-dialysis patients and dialysis patients on CCPB. Given that the calcium balance play a role in vascular calcification, could the titration of CCPB based on calcium-phosphate levels and the usage of non-calcium based phosphate binders play a role? This dynamic titration of phosphate binder doses and granular information may not be adequately captured in large electronic database studies.

6) Another important limitation for this study would be that for changes in prescribed doses of CCPB and the assurance of its continued use throughout the study period could not be assessed (some patients may be potentially converted to non-calcium based phosphate binder during the study due to hypercalcemia possibly?).

Minor comments

1) Abstract (results): it was written that 8124 patients were included but only 879 CCPB users were matched with 3516 nonusers. I think the first sentence may not be required. Clarifications need to be stipulated what were the actual patient population included in the 8124 patients in the abstract (methodology section) as it is not intuitive.

2) Abstract (results): The results regarding the mortality risk between CCPB users vs non-users should be described as no significant differences instead of lower but non-statistically significant.

3) Introduction: The authors have described the relationship between hyperphosphatemia and adverse outcomes extensively. It is also worthwhile to perform a short discussion of the role of hypercalcemia (given the study’s focus on calcium based phosphate binder) e.g. prevalence and related outcomes to better substantiate the need for this study. (consider citing: Int Urol Nephrol. 2018 Oct;50(10):1871-1877; J Clin Endocrinol Metab. 2016 Jun; 101(6): 2440–2449.)

4) What were the types of calcium based phosphate binder included in this study as they carry inherently different amount of elemental calcium?

5) Do consider defining “daily defined doses” in the methodology to help readers to understand this concept and its inherent limitations

Reviewer #3: The paper on Effects of Calcium-Containing Phosphate Binders on Cardiovascular Events and

Mortality in Predialysis CKD Stage 5 Patients support other observations that calcium load increase vascular calcification and in fact is non-traditional risk factor of cardiovascular event and death.

The manuscript is properly designed, conducted and analysed. The only points are listed below:

- many CKD4/5 patients take vitamin D analogs or supplement what may increase calcium absorpition. What percentage of patients did it in both groups?

- other drugs like Vitamin K antagonist increase independetly vascular calcification. How many patients took it?

Such a data are necesary for interpretation and is lacking must be commented as limitation of the study

Reviewer #4: This interesting study by Chu et al uses the Taiwanese national health insurance database to examine the association between use of calcium-containing phosphate binders (CCPB) and all-cause mortality and cardiovascular outcomes for CKD G5 patients. A few comments:

Abstract

-The methods section could include more details about the analysis - What kind of models were used in the analysis. How was cardiovascular events defined?

Introduction

-The focus and study population of the study could be clearer. There seems to be some inconsistency between the Introduction and the abstract as to whether this current study focuses on CKD stage 5 patients with or without hyperphosphatemia.

-Similarly, other studies have explored the link between calcium carbonate and adverse outcomes for pre-dialysis CKD stage 3 and 4. Why would the results be different for CKD stage 5. Please comment on the justification for using a CKD stage 5 population.

Methods

-How representative is the sample used in the study?

-It is not clear if number of admissions or any admissions (binary measure) was used as the outcome in models for coronary heart disease (line 134)?

-Was mortality assessed from the data in the Taiwan NHIRD or was there any data linkage to other sources (e.g. death registry)? If not, please comment on the completeness and accuracy of this data.

- CCPB users and non-users could also differ on medications and blood pressure and lab data. Were data on any of these available/looked at and why was this not included in the propensity score?

-Did the authors test for violations of the proportionality assumptions of the Cox regression models?

Results

-The authors found that adverse effects of CCPBs increased with increasing doses and cumulative high doses was also linked to higher risk of coronary heart disease. Were changes in the use of CCPB over the duration of the study period accounted for or explored in any way?

Discussion

-Good consideration of the limitations of the study.

6. PLOS authors have the option to publish the peer review history of their article (what does this mean?). If published, this will include your full peer review and any attached files.

Reviewer #1: No

Reviewer #2: No

Reviewer #3: No

Reviewer #4: **Yes: **Hilda Hounkpatin

---

## [Author Response · Author response to Decision Letter 0]

4 Sep 2020

Response to Reviewers:

We thank you for your time and effort to improve this study. The manuscript has been revised according to your valuable comments, and the changes in the revised manuscript of the marked-up version have been highlighted in red font. The itemized responses to four reviewers were listed as below.

Response to comments of Reviewer #1:

1. The authors did an excellent job analyzing the effect of calcium containing phosphate binders on cardiovascular mortality. However it is important that the authors identify with more detail the limitations of the study. The authors have mentioned the difficulty to separate Phosphate levels vs calcium load from the binder. There are other factors that should be commented: anticuoagulants-warfarin, treatment with active vitamin D, patients adherence to treatment not only relative to phosphtae control but also interdialytic weight gain, blood pressure medication, PTH levels , CRP levels and others.

Answer: Thank you very much for your comment. We have addressed other medication including warfarin or active vitamin D treatment as unmeasured confounders in the study limitations (line 307-308, page 18, marked-up version). Because of the limitation of NHIRD, the raw laboratory data were not included. We added CRP as an additional confounder (line 309, page 18, marked-up version). In this study, we enrolled predialysis CKD patients only, so we did not include interdialytic weight gain as a confounder.

Response to comments of Reviewer #2:

Major comments

1. It will be good for the authors to comment on their focus on Stage 5 predialysis patients and also why patients in stage 4 were not considered. [Noted this was only brought up in the discussion briefly but may be good to describe in the methodology]

Answer: Thank you very much for your comment. In this study, the identified information derived from the NHIRD included diagnostic codes according to the ICD9-CM; however, ICD-9-CM did not have a code define stage 1 to 4 CKD patients exactly. We enrolled the clinical condition of stage 5 CKD patients in the analyzed claims database by restricting patients to those receiving erythropoiesis-stimulating agents (ESAs), the medications that are reimbursed under the Taiwan universal health insurance program for patients whose serum creatinine concentrations are > 530 μmol/L. We described this section between line 105-110, page 5.

2. An important issue of note to evaluate the dose dependent relationship of calcium based phosphate binders is that the dose of elemental calcium within the CCB should be computed. This will facilitate more accurate and meaningful evaluation of this relationship as different CCPB have varying contents of elemental calcium. From the described methodology, it seems that it would be possible to compute the elemental calcium as the dose is available and the name of the CCPB.

Answer: We enrolled in calcium carbonate, calcium citrate, and calcium acetate in this study because these medications were provided by the national health insurance program in Taiwan. As the reviewer’s comment, different CCPB have varying contents of elemental calcium. In this study, we did not make a comparison among different CCPB directly; instead, we wanted to compare the effect of accumulated calcium content in CCPB on cardiovascular events. We used defined daily dose (DDD) as a technical unit of measurement defined as the assumed usual maintenance dose per day for a drug used for its main indication in adults. However, DDDs only give an estimate of consumption and not an exact picture of actual use. DDDs provide a fixed unit to assess trends in drug consumption and to perform comparisons between population groups.

3. How was the segregation of the subgroups of daily defined dose (≤15, 16–40, and >40 DDD) determined?

Answer: Thank you very much for your comment. DDD system allows standardization of drug groups and represents a stable drug utilization metric to enable comparisons of drug use and to examine trends in drug use over time. We decided to determine the subgroups of daily defined dose as tertile. 

4. An important factor for consideration that would substantially strengthen the relationship between use of CCPB and adverse outcomes would be the use of non-calcium based phosphate binders e.g. lanthanum, and sevalamer as well as vitamin D analogues. It will be good to report these information among users of CCPB and non-users and adjust for theses confounders (especially non-calcium based phosphate binders) in the multivariate analyses.

Answer: Under the Taiwan universal health insurance program, lanthanum and sevelamer were not covered by the insurance program, and these drugs are purchased out of patients’ pockets. Moreover, the medication at one’s own expense could not be validated in NHIRD. However, sevelamer was approved in 2003 and lanthanum was approved in 2006. Both of them were not popular phosphate binders in Taiwan during this study period (January 1, 2000, and June 30, 2005), which might mitigate the confounder effect of non-calcium phosphate binders. We addressed these medications including non-CCPB, warfarin, vitamin D as unmeasured variables in the study limitation (line 307-308, page 18, marked-up version).

5. Discussion: beyond the limits of study designs (comparisons made between winkelmayer et al), are there any potential reasons for the finding that there were no relationship between use of calcium based phosphate binders and mortality? In general most renal physicians will endeavour to avoid hypercalcemia among pre-dialysis patients and dialysis patients on CCPB. Given that the calcium balance play a role in vascular calcification, could the titration of CCPB based on calcium-phosphate levels and the usage of non-calcium based phosphate binders play a role? This dynamic titration of phosphate binder doses and granular information may not be adequately captured in large electronic database studies.

Answer: Thank you very much for your comment. Mineral metabolisms including calcium and phosphorus homeostasis were the promoter of vascular calcification among the CKD population. It is hard to maintain homeostasis of calcium and phosphorus levels among the advanced CKD population. Most renal physicians will prescribe CCPB first in Taiwan if hyperphosphatemia develops because CCPB is provided by the national health insurance program. Besides, during this study period (January 1, 2000, and June 30, 2005), lanthanum and sevelamer were not popular phosphate binders in Taiwan. We admitted that the titration of CCPB and usage of non-CCPB could have made the magnitude of the association of all-cause mortality toward the null. However, there is still no well-designed placebo-controlled trial demonstrating whether vascular calcification could be prevented or reversed with therapies aimed at maintaining calcium and phosphorus homeostasis in predialysis CKD patients. We think that this observational study still provides real-world data about the effect of CCPB use among predialysis CKD patients.

6. Another important limitation for this study would be that for changes in prescribed doses of CCPB and the assurance of its continued use throughout the study period could not be assessed (some patients may be potentially converted to non-calcium based phosphate binder during the study due to hypercalcemia possibly?).

Answer: Thank you very much for your comment. We agree that the misclassification of drug exposure of interest is possible, which could have resulted in the observed null association between CCPB use and all-cause mortality. We added a section to describe this important limitation (line 299-302, page 17-18, marked-up version).

Minor comments

1. Abstract (results): it was written that 8124 patients were included but only 879 CCPB users were matched with 3516 nonusers. I think the first sentence may not be required. Clarifications need to be stipulated what were the actual patient population included in the 8124 patients in the abstract (methodology section) as it is not intuitive.

Answer: Thank you very much for your comment. We had deleted the first sentence in line 29, page 2.

2. Abstract (results): The results regarding the mortality risk between CCPB users vs non-users should be described as no significant differences instead of lower but non-statistically significant.

Answer: Thank you very much for your comment. We had corrected as reviewer’s suggestion (line 33-34, page 2, marked-up version).

3. Introduction: The authors have described the relationship between hyperphosphatemia and adverse outcomes extensively. It is also worthwhile to perform a short discussion of the role of hypercalcemia (given the study’s focus on calcium based phosphate binder) e.g. prevalence and related outcomes to better substantiate the need for this study. (consider citing: Int Urol Nephrol. 2018 Oct;50(10):1871-1877; J Clin Endocrinol Metab. 2016 Jun; 101(6): 2440–2449.)

Answer: Thanks for your suggestion and we added a section to discuss the prevalence and adverse effect of hypercalcemia in predialysis CKD patients. Please see line 63-67, page 4, marked-up version.

4. What were the types of calcium based phosphate binder included in this study as they carry inherently different amount of elemental calcium?

Answer: Thank you very much for your comment. We enrolled in calcium carbonate, calcium citrate, and calcium acetate in this study because these medications were provided by the national health insurance program in Taiwan. These medication usages could be tracked from the registry for drug prescriptions in NHIRD. Considering the different amounts of elemental calcium, we used DDD to estimate and perform comparisons between population groups.

5. Do consider defining “daily defined doses” in the methodology to help readers to understand this concept and its inherent limitations.

Answer: Thank you very much for your comment. We added two sentences to describe the definition and inherent limitation of DDD between lines 158-162, page 9-10, marked-up version.

Response to comments of Reviewer #3:

1. many CKD4/5 patients take vitamin D analogs or supplement what may increase calcium absorpition. What percentage of patients did it in both groups?

Answer: Thank you very much for your comment. Use of vitamin D analogs or supplement is not provided by the national health insurance program in Taiwan. We could not track these medication usage data from the registry for drug prescriptions in NHIRD because these medications were at one’s own expense. We addressed vitamin D as unmeasured variables and one of the limitations (line 307-308, page 18, marked-up version).

2. other drugs like Vitamin K antagonist increase independetly vascular calcification. How many patients took it?

Answer: Thank you very much for your comment. We addressed Vitamin K antagonist and other drugs as unmeasured variables and one of the limitations (line 307-308, page 18, marked-up version).

Response to comments of Reviewer #4:

1. The methods section could include more details about the analysis - What kind of models were used in the analysis. How was cardiovascular events defined?

Answer: Thanks for your suggestion. We added a sentence to describe the statistical analysis in more detail (line 26-27, page 2, marked-up version). We described the statistical models for the main analysis in line 150-153, statistical analysis section, page 9. Considering the brief overview of the investigation, we decided to put the definition of cardiovascular events in line 141-142, page 9, marked-up version.

2. The focus and study population of the study could be clearer. There seems to be some inconsistency between the Introduction and the abstract as to whether this current study focuses on CKD stage 5 patients with or without hyperphosphatemia.

Answer: Thank you very much for your comment. Detailed laboratory test results and medical notes were not included in NHIRD, and baseline phosphorus, calcium, CRP, and parathyroid hormone levels were uncertain. Considering this limitation, we defined the inclusion of stage 5 CKD patients in the analyzed claims database by restricting patients to those receiving erythropoiesis-stimulating agents (ESAs), the medications that are reimbursed under the Taiwan universal health insurance program for patients whose serum creatinine concentrations are > 530 μmol/L.

3. Similarly, other studies have explored the link between calcium carbonate and adverse outcomes for pre-dialysis CKD stage 3 and 4. Why would the results be different for CKD stage 5. Please comment on the justification for using a CKD stage 5 population.

Answer: The different results of calcium use and adverse outcomes might be due to the diverse stage of the CKD population. Here are the reasons for using CKD stage 5 patients as the study population. First, hyperphosphatemia develops gradually in CKD patients with estimated glomerular filtration < 20 ml/min/1.73m. Second, advanced CKD patients have worse mineral metabolisms including calcium and phosphorus homeostasis, which were the promoter of vascular calcification among the CKD population.

4. How representative is the sample used in the study?

Answer: National health insurance covers the medical needs of 99.19% of the 23 million individuals in the population of Taiwan and the NHIRD comprises the standard health care data submitted by medical institutions, which is ideal for nationwide population-based cohort study.

5. It is not clear if number of admissions or any admissions (binary measure) was used as the outcome in models for coronary heart disease (line 134)?

Answer: Thanks for your comment. We used Cox model and the outcome was time to first event. If patients had multiple episodes of coronary heart disease, we analyzed the first event. NHIRD is a valid resource for population research on cardiovascular diseases, as the validity of acute myocardial infarction (AMI) diagnosis coding in the NHIRD has been demonstrated (J Epidemiol. 2014;24(6):500-7.). As published previously (J Clin Oncol. 2017 Nov 10;35(32):3697-3705.; Int J Cardiol. 2013 Oct 3;168(3):2616-21.), we estimated the incident coronary heart disease (ICD-9 codes 410–414, excluding 412 and 414.1) by employing ICD-9 diagnosis codes in inpatient, outpatient, and emergency department records.

6. Was mortality assessed from the data in the Taiwan NHIRD or was there any data linkage to other sources (e.g. death registry)? If not, please comment on the completeness and accuracy of this data.

Answer: Thanks for your comment. We assessed the cause of death as the primary diagnosis of in-hospital death or the first-listed discharge diagnosis at the last hospitalization within the three months before death for out-of-hospital deaths.

7. CCPB users and non-users could also differ on medications and blood pressure and lab data. Were data on any of these available/looked at and why was this not included in the propensity score?

Answer: Thanks for your comment. One of the major limitations of NHIRD is that detailed laboratory test results and medical notes were not included, and whether baseline laboratory tests or other non-calcium content phosphate binders were comparable between the two groups remains uncertain. We addressed this limitation to line 305-310, page 18, marked-up version.

8. Did the authors test for violations of the proportionality assumptions of the Cox regression models?

Answer: Thank you very much for your comment. We used Omnibus Test for the Cox Model. Test event of coronary heart disease, -2 Log likelihood = 26,786.101，Omnibus test: P<0.001; test event of all-cause mortality, -2 Log likelihood = 6,195.208，Omnibus test: P<0.001

9. The authors found that adverse effects of CCPBs increased with increasing doses and cumulative high doses was also linked to higher risk of coronary heart disease. Were changes in the use of CCPB over the duration of the study period accounted for or explored in any way?

Answer: Thank you very much for your comment. Use and nonuse of CCPB were determined within 90 days after the index date (the first date of using ESA); however, the drug status was not ascertained during the follow-up period. We admitted that the misclassification of drug exposure of interest is possible, which could have resulted in the observed null association between CCPB use and all-cause mortality. We added a section to describe this important limitation (line 299-302, page 17-18, marked-up version).

10. Good consideration of the limitations of the study.

Answer: Thank you very much for your comment.

---

## [Decision Letter · Decision Letter 1]

16 Sep 2020

PONE-D-20-04452R1

Effects of Calcium-Containing Phosphate Binders on Cardiovascular Events and Mortality in Predialysis CKD Stage 5 Patients

PLOS ONE

Dear Dr. Chu,

Thank you for submitting your manuscript to PLOS ONE. After careful consideration, we feel that it has merit but does not fully meet PLOS ONE’s publication criteria as it currently stands. Therefore, we invite you to submit a revised version of the manuscript that addresses the points raised during the review process.

There are still some comments that require the authors' attention. Please respond to ALL of reviewer #2's comments in your revised manuscript.

We look forward to receiving your revised manuscript.

Kind regards,

Frank T. Spradley

Academic Editor

PLOS ONE

Reviewers' comments:

Reviewer's Responses to Questions

**Comments to the Author**

1. If the authors have adequately addressed your comments raised in a previous round of review and you feel that this manuscript is now acceptable for publication, you may indicate that here to bypass the “Comments to the Author” section, enter your conflict of interest statement in the “Confidential to Editor” section, and submit your "Accept" recommendation.

Reviewer #2: (No Response)

Reviewer #4: All comments have been addressed

2. Is the manuscript technically sound, and do the data support the conclusions?

Reviewer #2: Yes

Reviewer #4: Yes

3. Has the statistical analysis been performed appropriately and rigorously? 

Reviewer #2: Yes

Reviewer #4: Yes

4. Have the authors made all data underlying the findings in their manuscript fully available?

Reviewer #2: No

Reviewer #4: Yes

5. Is the manuscript presented in an intelligible fashion and written in standard English?

Reviewer #2: Yes

Reviewer #4: Yes

6. Review Comments to the Author

Reviewer #2: Dear editor,

The authors have performed significant efforts to revise the manuscript based on the 4 reviewers' comments. One issue remained however inadequately addressed.

1. The computation of daily elemental calcium received from phosphate binders would actually permit a more meaningful comparison of calcium exposure compared to ddd as it reflects a more accurate assessment of calcium exposure to patient and a more standardised unit of measure. There is probably no need to compare between different phosphate binders as highlighted by the authors if they were to compute these as the unit of measure would be standardised across patients.

The authors may wish to comment on the above.

Thank you

Reviewer #4: My earlier comments and queries have been addressed. The methodology is appropriate and sound and the paper is presented clearly.

7. PLOS authors have the option to publish the peer review history of their article (what does this mean?). If published, this will include your full peer review and any attached files.

Reviewer #2: No

Reviewer #4: No

---

## [Author Response · Author response to Decision Letter 1]

12 Oct 2020

Response to Reviewer:

We thank you for your time and effort to improve this study. The manuscript has been revised according to your valuable comments, and the changes in the revised manuscript of the marked-up version have been highlighted in red font. The response to reviewer #2 was listed as below.

Response to comments of Reviewer #2:

1. The authors have performed significant efforts to revise the manuscript based on the 4 reviewers' comments. One issue remained however inadequately addressed. The computation of daily elemental calcium received from phosphate binders would actually permit a more meaningful comparison of calcium exposure compared to ddd as it reflects a more accurate assessment of calcium exposure to patient and a more standardised unit of measure. There is probably no need to compare between different phosphate binders as highlighted by the authors if they were to compute these as the unit of measure would be standardised across patients.

Answer: Thank you very much for your comment. We agreed with your comment that the computation of daily phosphate binders elemental calcium would permit more a meaningful comparison of calcium exposure compared to the defined daily dose because of a more accurate assessment of calcium exposure to the patient and a more standardized unit of measure. However, data in the National Health Insurance Research Database were de-identified by scrambling the identification codes of both patients and medical facilities, we could not track each patients’ accurate element calcium exposure or query the data at any level using this database. We added a section to describe this important limitation (line 301-303, page 18, marked-up version). To compare the effect of accumulated calcium content in CCPB on cardiovascular events, we used the defined daily dose (DDD) as a technical unit of measurement defined as the assumed usual maintenance dose per day for a drug used for its main indication in adults.

---

## [Editor Report · Decision Letter 2]

15 Oct 2020

Effects of Calcium-Containing Phosphate Binders on Cardiovascular Events and Mortality in Predialysis CKD Stage 5 Patients

PONE-D-20-04452R2

Dear Dr. Chu,

We’re pleased to inform you that your manuscript has been judged scientifically suitable for publication and will be formally accepted for publication once it meets all outstanding technical requirements.

Kind regards,

Frank T. Spradley

Academic Editor

PLOS ONE

---

## [Editor Report · Acceptance letter]

20 Oct 2020

PONE-D-20-04452R2 

Effects of Calcium-Containing Phosphate Binders on Cardiovascular Events and Mortality in Predialysis CKD Stage 5 Patients 

Dear Dr. Chu:

I'm pleased to inform you that your manuscript has been deemed suitable for publication in PLOS ONE. Congratulations! Your manuscript is now with our production department. 

Kind regards, 

on behalf of

Dr. Frank T. Spradley 

Academic Editor

PLOS ONE